# Near-Infrared Molecular Imaging of Glioblastoma by Miltuximab^®^-IRDye800CW as a Potential Tool for Fluorescence-Guided Surgery

**DOI:** 10.3390/cancers12040984

**Published:** 2020-04-16

**Authors:** Dmitry M. Polikarpov, Douglas H. Campbell, Lucinda S. McRobb, Jiehua Wu, Maria E. Lund, Yanling Lu, Sergey M. Deyev, Andrew S. Davidson, Bradley J. Walsh, Andrei V. Zvyagin, David A. Gillatt

**Affiliations:** 1Department of Clinical Medicine, Faculty of Medicine, Health and Human Sciences, Macquarie University, Sydney, NSW 2109, Australia; lucinda.mcrobb@mq.edu.au (L.S.M.); andrew.davidson@mq.edu.au (A.S.D.); david.gillatt@mq.edu.au (D.A.G.); 2Glytherix Ltd., Sydney, NSW 2113, Australia; douglas.campbell@glytherix.com (D.H.C.); angela.wu@minomic.com (J.W.); maria.lund@glytherix.com (M.E.L.); yanling.lu@glytherix.com (Y.L.); brad.walsh@glytherix.com (B.J.W.); 3Shemyakin-Ovchinnikov Institute of Bioorganic Chemistry RAS, 117997 Moscow, Russia; biomem@mail.ru; 4ARC Centre of Excellence for Nanoscale BioPhotonics, Macquarie University, Sydney, NSW 2109, Australia; 5Institute of Molecular Medicine, Sechenov University, 119991 Moscow, Russia

**Keywords:** brain neoplasm, fluorescence-guided surgery, glypican-1, IRDye800CW, Miltuximab, monoclonal antibodies, molecular imaging

## Abstract

Glioblastoma (GBM) is one of the most aggressive tumors and its 5-year survival is approximately 5%. Fluorescence-guided surgery (FGS) improves the extent of resection and leads to better prognosis. Molecular near-infrared (NIR) imaging appears to outperform conventional FGS, however, novel molecular targets need to be identified in GBM. Proteoglycan glypican-1 (GPC-1) is believed to be such a target as it is highly expressed in GBM and is associated with poor prognosis. We hypothesize that an anti-GPC-1 antibody, Miltuximab^®^, conjugated with the NIR dye, IRDye800CW (IR800), can specifically accumulate in a GBM xenograft and provide high-contrast in vivo fluorescent imaging in rodents following systemic administration. Miltuximab^®^ was conjugated with IR800 and intravenously administered to BALB/c nude mice bearing a subcutaneous U-87 GBM hind leg xenograft. Specific accumulation of Miltuximab^®^-IR800 in subcutaneous xenograft tumor was detected 24 h later using an in vivo fluorescence imager. The conjugate did not cause any adverse events in mice and caused strong fluorescence of the tumor with tumor-to-background ratio (TBR) reaching 10.1 ± 2.8. The average TBR over the 10-day period was 5.8 ± 0.6 in mice injected with Miltuximab^®^-IR800 versus 2.4 ± 0.1 for the control group injected with IgG-IR800 (*p* = 0.001). Ex vivo assessment of Miltuximab^®^-IR800 biodistribution confirmed its highly specific accumulation in the tumor. The results of this study confirm that Miltuximab^®^-IR800 holds promise for intraoperative fluorescence molecular imaging of GBM and warrants further studies.

## 1. Introduction

Glioblastoma is a malignant astrocytoma and the most common primary tumor of the brain [1]. Despite surgery and adjuvant therapy, GBM has one of the worst prognoses of all cancers with the median survival of only 12–15 months after diagnosis and a 5-year survival of approximately 5% [2,3,4]. While most of the radiologically defined tumor can often be surgically resected, the infiltrative nature of this disease does not allow good visualization and complete removal of the margins between invasive tumor and radiologically normal brain [5]. Several studies have demonstrated that the extent of the initial resection correlates with better patient outcomes [3,5,6,7,8], which means that better intraoperative visualization has the potential to benefit GBM patients by improving the extent of resection [9,10,11].

Fluorescence-guided surgery (FGS) is a technique that involves the use of a fluorescent contrast agent to improve intraoperative visualization of a tumor. Passive dyes fluorescein [12], indocyanine green [13,14], and 5-aminolevulinic acid (5-ALA) have been studied in clinical trials, however, only 5-ALA has been proven to benefit GBM patients and approved in many countries, including the USA, Canada, the EU, Australia, and Japan [10,15,16,17]. Such treatment renders cancer cells fluorescent and improves visualization during surgery, however, is not without shortfalls. Its mechanism of action results in low specificity, while its short excitation wavelength causes strong autofluorescence of normal tissue and limits the imaging penetration depth [10]. Due to these limitations, there is a growing interest in development and preclinical evaluation of more specific and efficient intraoperative imaging modalities [10].

Potential way to overcome the limitations of passive dyes is targeting of molecules that are present only on malignant cells [18,19,20,21,22]. Monoclonal antibodies, peptides, and other targeting ligands can recognize such molecules and deliver fluorescent dyes specifically to cancer cells and not the normal tissue [19,23]. This concept was initially approached by conjugating conventional passive dyes, such as fluorescein [24] and ICG [25], to targeting molecules but then the interest shifted towards more suitable fluorescent dyes which need be well tolerated, able to produce high-contrast image, and suitable for bioconjugation [22,26,27]. One of the dyes that fulfills these criteria and has shown the most promising results in clinical trials is IRDye800CW (IR800) [11,28]. This dye has an excellent safety profile and can be visualized by existing equipment developed for widespread FDA-approved dye indocyanine green, and its conjugates with anti-EGFR antibodies have been successfully used in preclinical and clinical trials for GBM and other cancers [28,29,30,31,32]. The conjugate of IR800 with anti-EGFR antibody cetuximab has been shown to access contrast-enhancing brain tumors and bind to glioma cells expressing target antigen without accumulation in the normal brain tissue in mice [30] and patients [28]. The lack of significant accumulation of such IR800- antibody conjugates in the brain has been also shown in mice [33] and *Cynomolgus macaques* [32]. Up to 60% of GBMs do not overexpress EGFR [34], and have very heterogenous antigen expression [35], so there is a need for other target antigens that can be used alone or in combination with EGFR.

A promising GBM oncotarget is a proteoglycan GPC-1, which is overexpressed in several cancers, including high- and low-grade gliomas and is associated with poor prognosis and resistance to therapy [36,37,38,39,40]. The presence of GPC-1 in aggressive GBM is in line with the known role of GPC-1 in tumor angiogenesis and metastasis [39], making it an attractive target for molecular imaging and FGS of actively growing GBM regions. Importantly, the expression of GPC-1 is limited to malignant tissue, as several studies have demonstrated its absence in normal astrocytes [36] and most adult normal tissue [41,42]. GPC-1 is not required for normal homeostasis, thus, targeting GPC-1 is likely to be safe [43].

Miltuximab^®^ is a chimeric antibody engineered from the parental antibody BLCA-38 [44] that targets GPC-1 [40]. We hypothesize that IR800 conjugated to Miltuximab^®^ can be used for intra-operative fluorescence imaging of GBM. Similarly to cetuximab-IR800, Miltuximab^®^-IR800 is expected to accumulate in contrast-enhancing brain tumors with damaged blood-brain barrier without accumulation in healthy brain tissue. The specificity and safety of Miltuximab^®^, recently demonstrated in a first-in-human clinical trial, make it an excellent candidate for intraoperative molecular imaging [45]. We synthesized and characterized the conjugate Miltuximab^®^-IR800 and investigated its application for fluorescence imaging of GBM in vitro and in vivo.

## 2. Results

### 2.1. Expression of Glypican-1 in Glioblastoma Cell Lines

The presence of GPC-1 on GBM cells U-251 and U-87 was confirmed by flow cytometry, compared to GPC-1-negative cell line Raji and analyzed using the murine version of Miltuximab^®^, antibody MIL-38 (GlyTherix Ltd., Sydney, Australia), and QIFIKIT (Quantitative Immunofluorescence Intensity kit, Dako) that allows quantification of surface-bound molecules. Flow cytometry demonstrated high expression of GPC-1 in both cell lines U-87 and U-251 (Figure 1) compared to the GPC-1 negative lymphoma cell line, Raji. Using a standard curve, U-87 cells were estimated to bind 10,395 anti-GPC-1 antibody molecules, while the value for the U-251 cell line was approximately 11-fold higher (116,941). The negative control cells Raji were estimated to bind approximately 261 antibody molecules. Lower expressing cell line U-87 was selected for the subsequent experiments.

### 2.2. Characterization of Miltuximab^®^-IR800

Chimeric monoclonal anti-GPC-1 IgG_1_ antibody Miltuximab^®^ was conjugated with an NHS ester of the NIR fluorescent dye IR800. The final concentration of the conjugate was found to be 0.57 mg/mL, and the dye/protein ratio was found to be 1:1 in both conjugates. The molecular purity was evaluated by size exclusion high-performance liquid chromatography (SEC) and found to be 90.2% monomeric for Miltuximab^®^ and 81.3% for Miltuximab^®^-IR800. Following the conjugation, the SEC data also demonstrated a shift of the main antibody peak and lower retention time, consistent with an increase in the size of the resultant molecule due to the attachment of the dye (Figure 2). Together, these findings are consistent with approximately 90% of monomeric Miltuximab^®^ molecules being dye labelled (and increased in size) and 10% of Miltuximab^®^ molecules remaining unlabeled and detected as a lower-molecular-weight fraction of Miltuximab^®^-IR800 at Figure 2. Similarly, an increase in size was observed for the control IgG-IR800 by SEC.

The binding of Miltuximab^®^-IR800 to GBM cells U-87 was characterized by flow cytometry and found to be comparable to that of unconjugated Miltuximab^®^ (Figure 3, left) confirming retention of GPC-1 binding capacity following conjugation. At the same time, the isotype control IgG-IR800 demonstrated almost no binding to U-87 cells, as expected (Figure 3, right).

### 2.3. In Vivo Fluorescence Imaging

Miltuximab^®^-IR800 or IgG-IR800 were intravenously injected into two groups of four immunodeficient mice bearing a subcutaneous GBM xenograft U-87 on the left hind leg. The U-87 cell line was selected for the in vivo study due to having lower GPC-1 expression than U-251. The mice were then imaged by a fluorescent imager daily for 10 days and sacrificed for ex vivo imaging of the tumors and major organs. The accumulation of Miltuximab^®^-IR800 was detectable in the tumor from day 1 and characterized by TBR = 3.7 ± 0.6 (Figure 4). The high specificity of the antibody was notable alongside very low background autofluorescence. By observation, the conjugate was gradually cleared from the circulation, but, importantly, retained in the tumor over the following 9 days, while the background fluorescence was gradually diminished. As a result, TBR of U-87 tumors peaked at 10.1 ± 2.8 and was significantly greater than that of the control conjugate (TBR of 1.7 ± 0.48; *p* = 0.029). To assess the overall performance of Miltuximab^®^-IR800 in fluorescence imaging, average fluorescence intensity for the 10-day imaging period was calculated and compared between the experimental groups. The average TBR for Miltuximab^®^-IR800 was significantly higher (5.8 ± 0.6) as compared to the IgG-IR800 cohort (2.4 ± 0.1; *p* = 0.001). Importantly, Miltuximab^®^-IR800 was well tolerated with no adverse effects observed. Body weights were stable throughout the experiment indicating that the treatment was well tolerated.

### 2.4. Ex Vivo Fluorescence of the Tumors and Major Organs

The experimental animals were sacrificed on day 10, and their tumor, liver, kidneys, spleen, and heart were collected, imaged, and analyzed. As evident from the ex vivo fluorescence images (Figure 5a), the fluorescence of the tumor was greater in the mice injected with Miltuximab^®^-IR800 in comparison with the organs or control tumors.

In order to mitigate the whole-organ fluorescence imaging artefacts to accurately quantify conjugate uptake, the organs were homogenized and reimaged. As presented in Figure 5b, the results were comparable to the ex vivo imaging of the whole organs with the highest accumulation of the dye in the tumors of the mice injected with Miltuximab^®^-IR800, as compared to the isotype control IgG-IR800 (*p* = 0.0025). Moreover, there was significantly higher accumulation of Miltuximab^®^-IR800 in the tumor when compared to the liver (*p* = 0.009) or kidneys (*p* = 0.011). We noted accumulation of systemically delivered monoclonal antibody in the liver for both Miltuximab^®^-IR800 and IgG-IR800 isotype control, as expected [46]. In the isotype control IgG-IR800, the uptake of the conjugates was found to be greater in the liver and kidney than in the tumor. Collectively, these data demonstrated that Miltuximab^®^-IR800 specifically accumulated in the U-87 tumor.

Frozen 5-µm-thick sections were prepared from the tumors and imaged to quantify their fluorescence. Figure 6 shows that the average mean fluorescence intensity of a tumor section from the mice injected with Miltuximab^®^-IR800 is significantly higher (30.65 ± 4.00 a.u.) than the average fluorescence in the control group tumor sections (9.42 ± 0.81 a.u.) (*p* = 0.005).

## 3. Discussion

Here, we have demonstrated the functionality of GPC-1 targeting antibody Miltuximab^®^-IR800 for molecular imaging of GBM. The high specificity of Miltuximab^®^-IR800 (and hence the promise of this conjugate as a potential tool for FGS) was confirmed using GBM cell lines and a GBM xenograft, demonstrating rapid accumulation and then retention in the tumor.

Fluorescence-guided surgery relies on the accumulation of fluorescent dye in the tumor, rendering it fluorescent and enabling intraoperative imaging. Visualization of the tumor is enabled by three mechanisms: (1) passive accumulation of intravenously delivered dye due to enhanced permeability of the tumor vasculature, (2) accumulation of dye in the tumor due to differences in metabolism of cancer cells, and (3) targeting of tumor cells by molecular imaging, which involves fluorescent labelling of specific target molecules present only on cancer cells, often using antibodies [10].

Passive accumulation of molecules in a GBM tumor occurs due to the loss of integrity of the blood-brain barrier, which was described by Swedish neurologist Tore Broman in 1944 [47]. Following his discovery, a group of researchers from the University of Minnesota demonstrated in 1948 that the fluorescent molecule, fluorescein, can be used for visualization of brain tumors [48]. Fluorescein and indocyanine green have been shown to accumulate in tumors in preclinical and clinical studies, however, their use has not been approved due to side-effects and low efficacy.

Fluorescence-guided surgery with 5-aminolevulinic acid (5-ALA) exploits the metabolic features of cancer cells. The molecule 5-ALA is necessary for the synthesis of heme and is a precursor of the natural fluorophore protoporphyrin IX. Excessive protoporphyrin IX can be converted by normal cells, however, it accumulates in cancer cells causing them to fluoresce. It has been studied more extensively than the passive agents and has been approved for use in GBM patients [16]. In 2006, Stummer et al. [49]. described the results of the first randomized controlled trial that demonstrated improved extent of resection and progression-free survival in patients treated by FGS with 5-ALA. Despite promising early results, 5-ALA has several shortfalls limiting its application. First, the nonspecific mechanism of 5-ALA imaging affects its utility as a tumor visualization agent. It has been described to cause fluorescence of nonmalignant tissue in cases of radiation necrosis or neurodegenerative disease [50]. Second, the excitation and emission of 5-ALA fall in the visible spectral range characterized by intense background autofluorescence of normal tissue eclipsing 5-ALA fluorescence [51].

Near-infrared molecular imaging has the potential to overcome both the limitations of 5-ALA. The agents designed for molecular imaging usually consist of a targeting component (cancer-specific antibody or antibody fragment) and an imaging component (fluorophore). The NIR fluorophore IR800 affords centimeter-deep optical imaging penetration depth on the background of diminished autofluorescence and as such, is more advantageous in comparison with ultraviolet–visible fluorophores such as 5-ALA or fluorescein [52]. The presence of a targeting molecule has the potential to provide better specificity than passive accumulation or 5-ALA. Indocyanine green has been investigated as a dye for molecular imaging. It was conjugated with chlorotoxin, which is a peptide that binds annexin A2, often present in malignant tissue (29). The ICG–chlorotoxin conjugate was termed tozuleristide or BLZ-100 and, after preclinical validation (30–32), it was tested in patients with glioma (15). The imaging agent BLZ-100 has shown promising results with specific accumulation in malignant tissue in most patients and reported no adverse events (15).

The current study sought to evaluate the utility of Miltuximab^®^-IR800, a chimeric monoclonal anti-GPC-1 antibody conjugated to IR800, as an agent for NIR molecular imaging of GBM. We first demonstrated binding of Miltuximab^®^ to GBM cell lines (U-87 and U-251) which was specific (as compared to an isotype control antibody). We then conjugated Miltuximab^®^ with IR800, and confirmed successful conjugation by spectrophotometry and SEC. The retention of GPC-1 binding by the conjugated antibody and fluorescence of the dye after conjugation were confirmed by flow cytometry. To assess the potential of Miltuximab^®^-IR800 as a specific tumor targeting agent when delivered systemically, it was intravenously administrated to mice bearing subcutaneous GBM tumors. In vivo imaging demonstrated specific accumulation of Miltuximab^®^-IR800 in the tumor which was apparent as soon as 24 h following delivery. Importantly, the antibody was retained in the tumor out to 10 days postinjection, with increasing contrast between tumor and nontumor tissues, evident from both in vivo and ex vivo analyses. This indicated stability of fluorescent signal as well as retention of the conjugate in the tumor, critical characteristics for an imaging agent used for FGS in clinical settings. Monoclonal antibodies are exquisitely specific for target antigen, and, given the established expression of GPC-1 in malignancy, including GBM, and the lack of GPC-1 expression seen in normal tissue, molecular targeting using Miltuximab^®^ may afford a specificity advantage over the use of 5-ALA. Of course, access of systemically delivered drugs to the brain remains a challenge for drug delivery in GBM—5-ALA as a small molecule achieves this, and, while antibodies have been known to cross the blood-brain barrier in GBM patients, access may not be equivalent to 5-ALA—a question that must be addressed to establish the utility of Miltuximab^®^-IR800 for imaging of GBM.

Miltuximab^®^ represents an emerging, highly potent targeting agent suitable for rapid clinical translation. The safety of Miltuximab^®^ conjugated with a radioimaging isotope has been demonstrated in a Phase I first-in-human trial [45]. In line with what has been seen clinically, the safety of targeting GPC-1 has been shown in vivo in animal models. A preclinical study using an anti-GPC-1 antibody that cross-reacts with mouse GPC-1 demonstrated safety to 50 mg/kg dose of antibody delivered intravenously in immune-competent mice [41]. That study examined serum chemistry, blood counts, and histopathology on organs (liver, lung, heart, kidneys, brain, spleen, and testis) in mice treated with anti-GPC-1 antibody (50 mg/kg) or an isotype control antibody and found no pathology associated with targeting of GPC-1 [41]. In a similar study designed to assess safety, treatment of immune-competent mice with 50 mg/kg anti-GPC-1 cytotoxic antibody drug conjugate (ADC) did not reveal any safety signal [53].

The safety profile of IR800 and its conjugates have also been established. Its pharmacokinetics and safety were studied in animal models and then confirmed in clinical trials [11,28,29,30,31,32,54,55]. It has an excellent safety profile and has not been described to cause any significant adverse effects.

Our study confirms that the fluorescent conjugate Miltuximab^®^-IR800 can target GPC-1 to identify and visualize GBM tumors in vivo, with long retention of fluorescent signal in the tumor. The collective data on both Miltuximab^®^, IR800 dye and the Miltuximab^®^-IR800 conjugate, suggest that it has excellent safety. Thus, Miltuximab^®^-IR800 has high potential for clinical utility as an FGS imaging agent. The main limitations of this study are the use of a single cell line in vivo and a heterotopic, rather than an orthotopic, xenograft model. To confirm its utility, future studies will examine targeting of Miltuximab^®^-IR800 in other cell-line-based or patient-derived orthotopic xenograft or induced models of GBM and will evaluate improvement in tumor resection with Miltuximab^®^-IR800 using clinical imaging equipment as a further evaluation of the translational aspects of this technology. Such orthotopic models will allow further confirmation of the molecular specificity of the targeting, evaluation of the fluorescence contrast of tumor versus normal brain tissue, and determination of the extent of resection that can be achieved using Miltuximab^®^-IR800.

## 4. Materials and Methods

### 4.1. Cell Lines

GBM cell line U-87 MG (ATCC HTB-14) and lymphoma cells Raji (ATCC CCL-8) were sourced from ATCC (USA). GBM cell line U-251 MG was purchased from Merck (Germany). The origin of the cell lines and the absence of mycoplasma contamination were confirmed every 3 months by Cell Bank Australia. The cells were cultured according to the standard culturing protocol in a 5% CO_2_ tissue culture incubator at 37 °C. U-87 and U-251 cells were grown in low-glucose Dulbecco’s Modified Eagle’s Medium (DMEM; Sigma-Aldrich, St. Louis, MO, USA) supplemented with 1% nonessential amino acids (Sigma, USA) and 10% fetal bovine serum (Scientifix, Clayton, VIC, Australia). Raji cells were cultured in RPMI 1640 medium (Invitrogen, Carlsbad, CA, USA) supplemented with 10% heat inactivated (56 °C, 30 min) fetal bovine serum. All cell lines were cultured in T175 cell culture flasks (Greiner Bio-One, Frickenhausen, Germany). The cells were detached by rinsing with phosphate-buffered saline (PBS) and incubating with 2 mM ethylenediaminetetraacetic acid (EDTA) in PBS at 37 °C for 15–20 min.

### 4.2. Quantification of Glypican-1 Expression on the Surface of Glioblastoma Cells

The expression of GPC-1 by GBM cells was quantified using the murine version of Miltuximab^®^, antibody MIL-38 (GlyTherix Ltd., Sydney, Australia), and QIFIKIT (Quantitative Immunofluorescence Intensity kit, Dako) as per the manufacturer’s protocol. Briefly, the U-87, U-251, and Raji cells were incubated with an anti-GPC-1 antibody MIL-38 and then a secondary FITC-conjugated goat antimouse antibody from the kit. QIFIKIT beads with known number of antibody molecules on the surface were incubated with the same secondary FITC-conjugated goat antimouse antibody. Both cells and beads were analyzed by BD Fortessa X20 (BD Biosciences, Franklin Lakes, NJ, USA) flow cytometer. The fluorescence data from the beads were used to produce a calibration curve. The number of bound antibody molecules on the cell surface was then interpolated from the calibration curve.

### 4.3. Conjugation and Characterization of Miltuximab^®^-IR800 

Chimeric monoclonal anti-GPC-1 IgG1 antibody Miltuximab^®^ (GlyTherix Ltd., Australia) was conjugated with the NIR fluorescent dye IR800 via an N-Hydroxysuccinimide (NHS) ester (for lysine binding on the antibody) using an IRDye 800CW Protein Labeling Kit—high MW (928-33040, LI-Cor Biosciences, Lincoln, NE, USA) as per the manufacturer’s protocol. The Miltuximab^®^-IR800 conjugate was subsequently separated from the free dye by centrifugation at 1000 *× g* for 2 min in a Zeba Spin Desalting Column with 7 kDa molecular weight cut off (Thermo Fisher Scientific, Waltham, MA, USA), supplied in the kit. To determine the final concentration of the conjugate and the dye/antibody molar ratio, absorbance spectra at 280 nm (A_280_) and 780 nm (A_780_) were determined by a microplate reader PHERAstar (BMG Labtech, Ortenberg, Germany). Following the protocol supplied with the kit, the dye/antibody molar ratio was calculated as [A_780_/ε_IR800_]/[(A_280_ − (0.03 × A_780_))/ε_Antibody_], where 0.03 is a correction factor for the absorbance of IR800 at 280 nm, the molar extinction coefficient of the dye (ε_IR800_) is 270,000 M^−1^cm^−1^ and the molar coefficient of the protein (ε_Antibody_) is 203,000 M^−1^cm^−1^. The final concentration of the conjugate in mg/mL was calculated as [(A_280_ − (0.03 × A_780_))/ε_Antibody_] × MW_Antibody_ × dilution factor, where MW_Antibody_ is the antibody molecular weight. A human IgG Isotype Control (cat # 31154, Thermo Fisher Scientific, USA) was also conjugated with IR800 and the resulting control IgG-IR800 conjugate was characterized using the same method as for Miltuximab^®^-IR800.

The % purity of the antibody and conjugate samples was assessed by SEC using an Agilent Bio SEC-3 column on an Agilent 1200 HPLC system (Agilent Technologies, Inc., Santa Clara, CA, USA). Overall, 5 uL sample was injected via the autosampler of the HPLC system and chromatograph was recorded with UV detection at 280 nm, with flow rate of 0.3 mL/min for 20 mins and column temperature of 25 °C. A protein gel filtration standard mixture (Bio-Rad Cat#1511901, Hercules, CA, USA) was run as a system control prior to analysis of all other samples. Data processing was performed via Agilent ChemStation software system (Agilent Technologies, Inc., Santa Clara, CA, USA) which gave the % purity via integration of area under the curve of individual peak.

The specific binding of Miltuximab^®^-IR800 to U-87 cells was compared to the binding of unconjugated Miltuximab^®^ and control IgG-IR800 to the same cell line by flow cytometry. The U-87 cells (5 × 10^5^) were incubated with 50 µL Miltuximab^®^-IR800, Miltuximab^®^, or IgG-IR800 at a saturating concentration of antibody (10 µg/mL) for 45 min on ice, washed three times by PBS, and then, incubated for 30 min with 50 µL of 10 µg/mL of the fluorescently labelled secondary antibody Alexa Fluor 488 goat antihuman IgG (H + L) (Life Technologies, Carlsbad, CA, USA). The cells were then washed three more times with PBS and analyzed with a BD Fortessa X20 flow cytometer (BD Biosciences, Franklin Lakes, NJ, USA).

### 4.4. Establishment of a Subcutaneous Mouse Model of Glioblastoma

All applicable institutional and/or national guidelines for the care and use of animals were followed. The project was approved by the Macquarie University Animal Ethics Committee (AEC Reference No.: 2018/015) and carried out in accordance with the Australian code for the care and use of animals for scientific purposes [56]. For the subcutaneous inoculation, U-87 cells were suspended in serum-free DMEM media mixed 1:1 with Matrigel Basement Membrane matrix (Corning, Corning, NY, USA). Ten 8-week old female BALB/c nude mice (Animal Resource Centre, Canning Vale, Australia) were then anaesthetized by isoflurane (4% induction and 2% maintenance), and one million cells in 50 µL of medium were injected subcutaneously to the left hind leg. The tumor growth was assayed by caliper measurement. The mice were monitored daily for signs of distress and weighed twice weekly.

### 4.5. In Vivo Imaging of the Subcutaneous Tumors

When the tumors reached approximately 1000 mm^3^ in volume, eight mice were selected and randomized into two groups to be injected with Miltuximab^®^-IR800 or control isotype IgG-IR800. The conjugates were suspended in PBS at 0.6 mg/mL, filtered using a 0.22-µm syringe filter, and injected into the tail vein at 6 mg/kg in isoflurane-anesthetized animals. The mice were then imaged by an Odyssey CLx fluorescence imager (LI-Cor Biosciences, Lincoln, NE, USA) at 3, 10, and 24 h postinjection and then daily for 10 days. For the imaging, the mice were anesthetized by intraperitoneal injection of a combination of 20 mg/kg Zoletil (tiletamine hydrochloride and zolazepam hydrochloride; Virbac Pty Ltd., Milperra, Australia) and 0.5 mg/kg Domitor (medetomidine hydrochloride; Zoetis Australia Pty Ltd., Rhodes, Australia) with reversal by 0.5 mg/kg Antisedan (atipamezole; Zoetis Australia Pty Ltd., Rhodes, Australia). Image Studio software (LI-Cor Biosciences, Lincoln, NE, USA) was used to establish the TBR from the images. Fluorescence was compared between regions of interest of equal size placed on the tumor or on the back of the mouse adjacent to the tumor region.

### 4.6. Ex Vivo Imaging and Biodistribution Study

At 10 days after the intravenous injection of Miltuximab^®^-IR800 or IgG-IR800, the mice were sacrificed by isoflurane overdose with subsequent cervical dislocation. The tumors and major organs were then collected, imaged using the Odyssey CLx fluorescent imager, and preserved by freezing for subsequent analysis. In order to avoid the effect of possible attenuation of light on ex vivo imaging of whole tumors and organs of variable thickness, the samples were later homogenized and analyzed as was described by Oliveira et al. [57]. Briefly, a sample of 50–150 µg of each tumor and organ was collected and homogenized. For the homogenization, the samples were suspended in 800 µL RIPA buffer (50 mM Tris-HCl, pH 7.4, 150 mM NaCl, 1 mM EDTA, 1% Triton X-11, 0.1% SDS, and 0.5% sodium deoxycholate) in a tube with a 1/4 in. Ceramic Sphere and 1.4 mm Ceramic Beads (MP Biomedicals, Santa Ana, CA, USA). The samples were then disrupted by a FastPrep-24 homogenizer (MP Biomedicals Santa Ana, CA, USA) at 40 sec, 5 min off, and three times at room temperature. The samples were then serially diluted with RIPA buffer in a clear bottom 96-well plate, and imaged with the Odyssey CLx.

### 4.7. Ex Vivo Fluorescence Imaging of Sectioned Tumors

To further quantify and compare the fluorescence of the tumors from the mice injected with Miltuximab^®^-IR800 or IgG-IR800, the frozen tumors were cryosectioned and imaged ex vivo. Three 5-µm sections of each tumor were imaged with the Odyssey CLx, and mean fluorescence intensity of the sections was analyzed with the Image Studio software (LI-Cor Biosciences, Lincoln, NE, USA). The fluorescence intensity of the tumor sections was then compared between the groups.

### 4.8. Statistical Analysis

The values are expressed as mean ± standard error of the mean (SEM). The statistical analyses were performed using GraphPad Prism software. The in vivo TBR and ex vivo mean fluorescence intensity were compared between the groups using the unpaired student’s *t* test with Welch’s correction. A *p* value of less than 0.05 was considered statistically significant.

## 5. Conclusions

Fluorescence-guided surgery has been proven to benefit GBM patients through precise determination of tumor margins and improvement of the extent of resection. Molecular imaging of GBM-specific antigens appears to be the most promising method, where high-specificity antibodies are pivotal. Anti-GPC-1 antibody, Miltuximab^®^, appears to be a suitable molecule showing specific binding to GBM cells lines U-87 and U-251. For the first time, to the best of our knowledge, an anti-GPC-1 antibody was conjugated to an NIR dye IR800. Miltuximab^®^-IR800 enabled high-specificity high-contrast molecular imaging of GBM in vitro and in vivo in a xenograft model, supporting the clinical development of this conjugate as a tool for FGS. This study will pave the way for advanced FGS, able to provide improved discrimination between cancerous and normal tissues, and hence improve extent of resection.

## Figures and Tables

**Figure 1 cancers-12-00984-f001:**
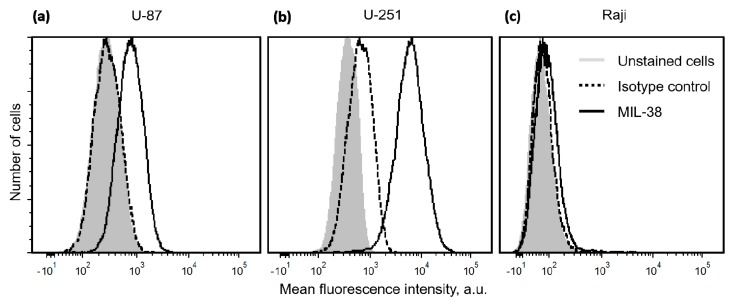
Histograms demonstrating flow cytometry analysis of the binding of anti-GPC-1 antibody MIL-38 to GBM cells U-87 (**a**), U-251 (**b**), and lymphoma cells Raji (**c**). The light gray filled histogram represents unstained cells, dashed black line shows the cells incubated with an isotype control antibody, and black line shows the cells incubated with MIL-38.

**Figure 2 cancers-12-00984-f002:**
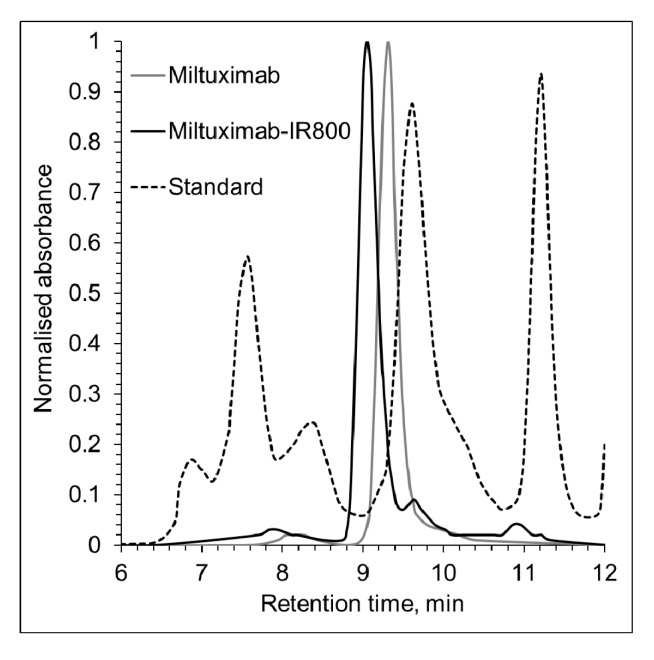
Size exclusion chromatography of Miltuximab^®^ and Miltuximab^®^-IR800. Miltuximab^®^-IR800 conjugate was tested for purity and conjugation efficiency by size-exclusion chromatography using Agilent Bio SEC-3 column tuned to UV detection at 280 nm. The conjugation of IR800 dye to Miltuximab^®^ produced a single peak with a shift to the left indicating a shorter retention time, consistent with an increase of the molecule size. A black dotted line represents the protein standard that shows the following peaks from left to right: thyroglobulin (670 kDa), y-globulin (158 kDa), and ovalbumin (44 kDa).

**Figure 3 cancers-12-00984-f003:**
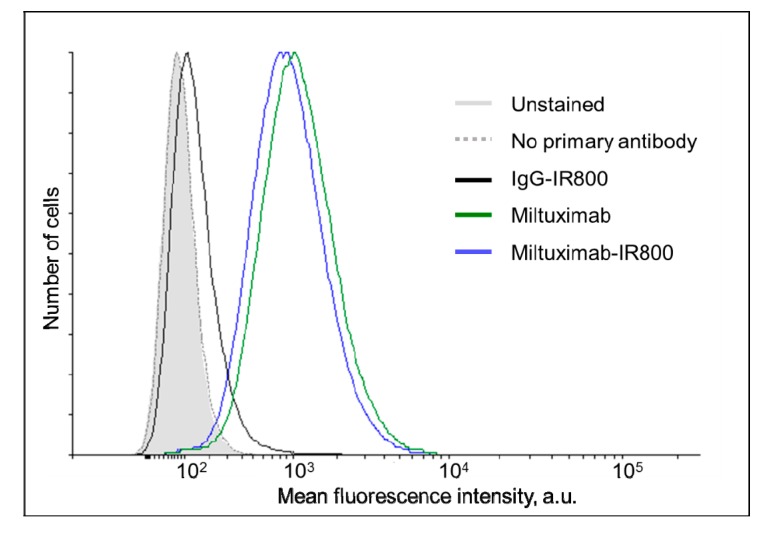
Histograms demonstrating the flow cytometry analysis of the labelling of U-87 cells by Miltuximab^®^-IR800 (solid black line), IgG-IR800 (dark gray line), or unconjugated Miltuximab^®^ (dotted black line). After incubation with the conjugates, the cells were stained by an anti-human-IgG antibody conjugated to a fluorescent dye AlexaFluor488. The light gray-filled histogram represents unstained cells and the dotted gray line shows the cells incubated only with the secondary antibody.

**Figure 4 cancers-12-00984-f004:**
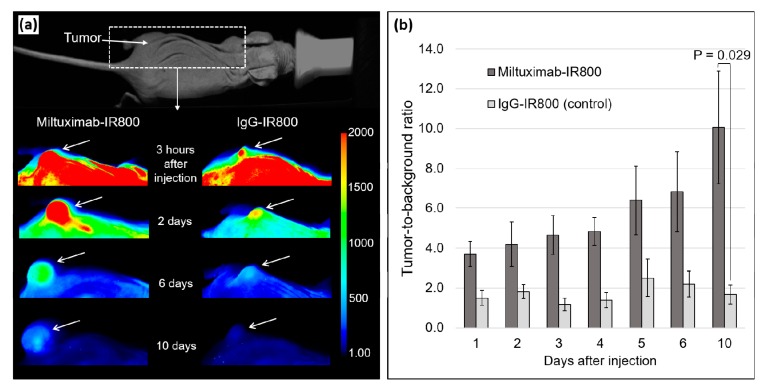
(**a**) False color fluorescent imaging of living mice injected with Miltuximab^®^-IR800 or IgG-IR800 at 3 h and 2, 6, and 10 days after the intravenous injection under the same imaging parameters, including min/max pixel values. White arrows indicate the location of the U-87 tumor. (**b**) Tumor-to-background ratios in U-87 tumors from in vivo fluorescent imaging in mice injected with Miltuximab^®^-IR800 (dark gray) or IgG-IR800 (light gray). The total fluorescence intensity was measured from the regions of interest of equal areas placed on tumors or mouse dorsal side as a background. Data represent mean fluorescence intensity ± SEM of three or four independent animals. The *p* value was determined by a *t* test with Welch’s correction.

**Figure 5 cancers-12-00984-f005:**
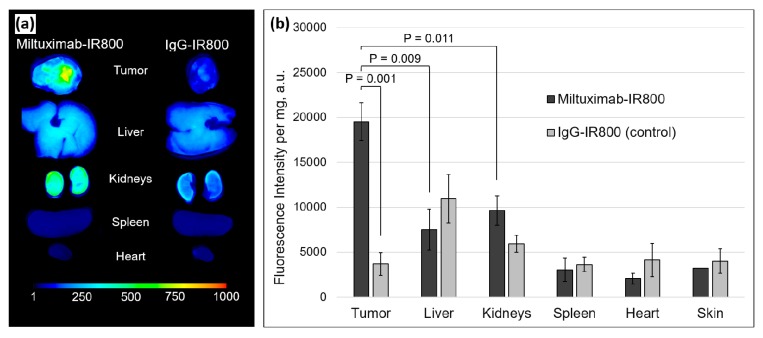
(**a**) Ex vivo fluorescence imaging of tumor, liver, kidneys, spleen, and heart removed from the mice 10 days after the intravenous injection of Miltuximab^®^-IR800 (left) or control IgG-IR800 (right). (**b**) Bar chart of the fluorescence intensity of the homogenized tumors and organs imaged in a 96-well plate. Dark and light gray colors mark Miltuximab^®^-IR800 and control samples, respectively. Data represent mean fluorescence intensity ± SEM of the samples from three or four independent animals. *p* values were determined by a Welch’s *t* test.

**Figure 6 cancers-12-00984-f006:**
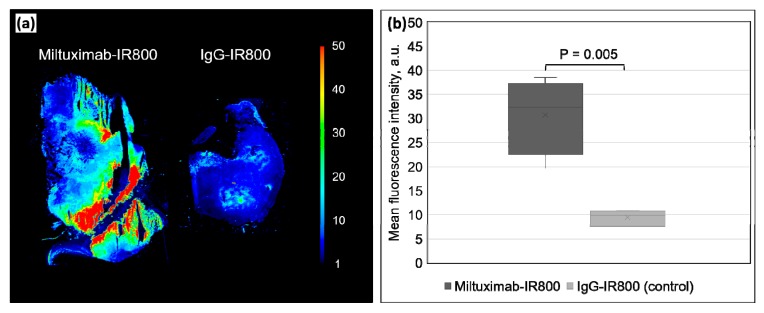
(**a**) A representative false-color fluorescence image of thin-sectioned specimens of U-87 tumor harvested from the mice injected with Miltuximab^®^-IR800 (left) or control IgG-IR800 (right). (**b**) Box plot of the fluorescence intensity of the tumor sections from the mice injected with Miltuximab^®^-IR800 (dark gray) or IgG-IR800 (light gray). Data represent mean fluorescence intensity of three sections ± SEM from three or four independent animals. Statistical significance and *p* values were determined by a Welch’s *t* test.

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
