# Peer review of "Near-Infrared Molecular Imaging of Glioblastoma by Miltuximab®-IRDye800CW as a Potential Tool for Fluorescence-Guided Surgery"

_cancers, 2020, doi:10.3390/cancers12040984_

Round 1
Reviewer 1 Report
Although in vivo and ex vivo detection of more kinds of glioblastoma cells and that in brain should be checked in order to show clinical usability of Miltuximab®-IR800, this manuscript shows the importance of glypican-1 expression in glioblastoma (and other cancers) and encourages future translational research to make use of Miltuximab®-IR800 in clinical imaging.
Author Response
Thank you for the comment, the authors believe that this work demonstrates the potential of Glypican-1 in fluorescence-guided surgery of glioblastoma and paves the way for further translational research.
Reviewer 2 Report
This is a very straightforward study. The authors conjugated Miltuximab with IR800 (Miltuximab-IR800). They intravenously administered Miltuximab-IR800 to Balb/c nude mice bearing a subcutaneous U-87 GBM hind leg xenograft and performed in vivo fluorescence imaging. They reported accumulation of Miltuximab-IR800 in subcutaneous xenograft tumor with high tumor-to-background ratio. The average TBR of Miltuximab-IR800 over the 10-day period was significantly higher than that of the control group injected with IgG-IR800. Ex vivo assessment revealed specific accumulation in the tumor. The authors claim that Miltuximab-IR800 holds high potential for clinical utility as an FGS imaging agent. However, the imaging data obtained in a subcutaneous GBM model did not directly support their claim. They need to discuss the limitations and pitfalls of their study. Minor: It is hard to see the color difference between the black and dark grey lines in Figure 3.
Author Response
Comment:
The authors claim that Miltuximab-IR800 holds high potential for clinical utility as an FGS imaging agent. However, the imaging data obtained in a subcutaneous GBM model did not directly support their claim. They need to discuss the limitations and pitfalls of their study.
Reply:
The main limitations of the study are the use of a single cell line in vivo and a heterotopic, but not orthotopic GBM model. An amendment has been made in the text to highlight this.
Amendment:
The last paragraph of the section “3. Discussion” (lines 291-294) has been revised and now includes the following:
“The main limitations of this study are the use of a single cell line in vivo and a heterotopic, rather than an orthotopic xenograft model. To confirm its utility, future studies will examine targeting of Miltuximab®-IR800 in other cell-line-based or patient-derived orthotopic xenograft or induced models of GBM and will evaluate the improvement in tumor resection with Miltuximab®-IR800 by using clinical imaging equipment as a further evaluation of the translational aspects of this technology.”
Comment:
Minor: It is hard to see the color difference between the black and dark grey lines in Figure 3.
Reply:
The color of the lines in Figure 3 has been changed to provide better contrast.
Reviewer 3 Report
The authors present the paper “Near-infrared molecular imaging of glioblastoma by Miltuximab-IRDye800CW as a potential tool for fluorescence-guided surgery.” In this article, they assessed the potential benefit of using Miltuximab-IR800 for intraoperative fluorescence molecular imaging of GBM using an in-vitro and in-vivo model. This is a well written comprehensive paper that will be well received. My comments to the authors are as follows:
1.Introduction
Line 47: The authors discuss the shortfalls of using 5-ALA for fluorescence-guided surgery. Other agents such as BLZ-100 (a CTX-indocyanine green conjugate) are being used in pre-clinical and clinical trials for adults and children with brain tumors. Please expand on this section (and/or the similar section in the discussion) to include additional agents other than just 5-ALA that have been studied in pre-clinical models and are currently being incorporated into clinical trials.
Line 65: Please check the spelling of “1R800-antivbody”
2. Results
Please comment on why only the U-87 cell line was used for the majority of the in-vitro and in-vivo studies and not both the U-87 and U-251 cell lines, as both lines have high GPC-1 expression. It would have been nice to see the results validated with a second cell line/tumor model.
It is unclear how many mice were used for each experiment. Please include the actual number of animals used for each experiment, rather than “at least 3 independent animals.”
Section 2.4 line 157: Tumor, liver, kidneys, spleen and heart were collected and analyzed. This paper is proposing the use of Miltuximab-IRDye800CW for a brain tumor. Although the authors state that future studies will be done with orthotopic or xenograft models of GBM, it would be nice if the authors included fluorescence values in the brains of mice injected with the Miltuximab-IRDye800CW conjugate as well as other organs
Author Response
Comment:
Line 47: The authors discuss the shortfalls of using 5-ALA for fluorescence-guided surgery. Other agents such as BLZ-100 (a CTX-indocyanine green conjugate) are being used in pre-clinical and clinical trials for adults and children with brain tumors. Please expand on this section (and/or the similar section in the discussion) to include additional agents other than just 5-ALA that have been studied in pre-clinical models and are currently being incorporated into clinical trials.
Reply:
The authors have added more information about other agents for fluorescence-guided surgery of glioblastoma in Introduction and Discussion.
Amendment:
The Introduction now includes the following:
“Passive dyes fluorescein [12], indocyanine green [13,14] and 5-aminolevulinic acid (5-ALA) have been studied in clinical trials, however, only 5-ALA has been proven to benefit GBM patients and approved in many countries, including the USA, Canada, the EU, Australia and Japan [10,15–17].” (lines 48-51)
“Potential way to overcome the limitations of passive dyes is targeting of molecules that are present only on malignant cells [18–22]. Monoclonal antibodies, peptides and other targeting ligands can recognize such molecules and deliver fluorescent dyes specifically to cancer cells and not normal tissue [19,23]. This concept was initially approached by conjugating conventional passive dyes, such as fluorescein [24] and ICG [25], to targeting molecules but then the interest shifted towards more suitable fluorescent dyes which need be well tolerated, able to produce high-contrast image, and suitable for bioconjugation [22,26,27]. One of the dyes that fulfills these criteria and has shown the most promising results in clinical trials is IRDye800CW (IR800) [11,28].” (lines 58-65)
The following text has been incorporated into the Discussion (line 239-244):
“Indocyanine green has been investigated as a dye for molecular imaging. It was conjugated with chlorotoxin, which is a peptide that binds annexin A2, often present in malignant tissue (29). The ICG-chlorotoxin conjugate was termed tozuleristide or BLZ-100 and, after preclinical validation (30–32), was tested in patients with glioma (15). The imaging agent BLZ-100 has shown promising results with specific accumulation in malignant tissue in most patients and no reported adverse events (15).”
Comment:
Line 65: Please check the spelling of “1R800-antivbody”
Reply and amendment:
Thank you, the spelling has been corrected and now says “IR800-antibody”.
- Results
Comment:
Please comment on why only the U-87 cell line was used for the majority of the in-vitro and in-vivo studies and not both the U-87 and U-251 cell lines, as both lines have high GPC-1 expression. It would have been nice to see the results validated with a second cell line/tumor model.
Reply:
The accumulation of IR800-antibody conjugate in both U-87 and U-251 xenografts in vivo has been reported before [4]. We demonstrated Glypican-1 expression in both U-87 and U-251 cell lines and selected the lower-expressing U-87 cells for the animal model. U-251 cells have significantly higher expression of Glypican-1 and therefore are unlikely to have less accumulation of Miltuximab®-IR800.
Amendment:
The sentence: “Lower-expressing cell line U-87 was selected for subsequent experiments.” Was added to the section “2.1. Expression of Glypican-1 in glioblastoma cell lines” (line 105).
The sentence “The U-87 cell line was selected for the in vivo study due to having lower GPC-1 expression than U-251.” Has been added to the section 2.3. In vivo fluorescence imaging, lines 144-145.
Comment:
It is unclear how many mice were used for each experiment. Please include the actual number of animals used for each experiment, rather than “at least 3 independent animals.”
Reply:
The total of 8 mice were randomised into 2 groups of 4 and injected with Mituximab-IR800 or control IgG-IR800. At several time points only 3 mice could be imaged per group.
Amendment:
The first sentence of section 2.3. In vivo fluorescence imaging (lines 143-144) now reads “injected into 2 groups of 4 immunodeficient mice” instead of “injected into immunodeficient mice”
2 groups of 4
The words “at least 3 independent animals” have been replaced with “3 or 4 independent animals” in the caption of figures 4, 5, and 6.
Comment:
Section 2.4 line 157: Tumor, liver, kidneys, spleen and heart were collected and analyzed. This paper is proposing the use of Miltuximab-IRDye800CW for a brain tumor. Although the authors state that future studies will be done with orthotopic or xenograft models of GBM, it would be nice if the authors included fluorescence values in the brains of mice injected with the Miltuximab-IRDye800CW conjugate as well as other organs
Reply:
No fluorescence signal from the brain was detected in vivo. Accumulation of Miltuximab®-IR800 in healthy brain is unlikely due to the intact blood-brain barrier (BBB), and hence no images of the brain post mortem were collected from the animals. The lack of significant non-specific accumulation of Nimotuzumab-IR800 and control IgG-IR800 in the mouse brain has been reported [1], and no significant specific accumulation of Cetuximab-IR800 was found in the Cynomolgus Macaque brain [2]. The accumulation of the conjugate in the brain was significantly lower than that in the other critical organs.
On the other hand, Miltuximab®-IR800 is expected to accumulate in the brain tumours due to damaged blood-brain barrier and remain there due to the binding to Glypican-1. The conjugates of IR800 with integrin-targeting peptide or Cetuximab have been shown to gain access the orthotopic glioblastoma xenografts in mice [3] with no accumulation in the normal brain tissue. The accumulation of Cetuximab-IR800 in contrast-enhancing glioma was later demonstrated in patients [4]. Similarly to Cetuximab, Miltuximab® is a chimeric mouse/human IgG1 antibody. Miltuximab®-IR800 is therefore expected to behave similarly in vivo with accumulation in contrast-enhancing glioma, but not in normal brain tissue.
- Bernhard, W.; El-Sayed, A.; Barreto, K.; Gonzalez, C.; Hill, W.; Parada, A.C.; Fonge, H.; Geyer, C.R. Near infrared fluorescence imaging of EGFR expression in vivo using IRDye800CW-nimotuzumab. Oncotarget 2018.
- Zinn, K.R.; Korb, M.; Samuel, S.; Warram, J.M.; Dion, D.; Killingsworth, C.; Fan, J.; Schoeb, T.; Strong, T. V.; Rosenthal, E.L. IND-Directed Safety and Biodistribution Study of Intravenously Injected Cetuximab-IRDye800 in Cynomolgus Macaques. Imaging Biol. 2014.
- Warram, J.M.; De Boer, E.; Korb, M.; Hartman, Y.; Kovar, J.; Markert, J.M.; Gillespie, G.Y.; Rosenthal, E.L. Fluorescence-guided resection of experimental malignant glioma using cetuximab-IRDye 800CW. J. Neurosurg. 2015.
- Miller, S.E.; Tummers, W.S.; Teraphongphom, N.; van den Berg, N.S.; Hasan, A.; Ertsey, R.D.; Nagpal, S.; Recht, L.D.; Plowey, E.D.; Vogel, H.; et al. First-in-human intraoperative near-infrared fluorescence imaging of glioblastoma using cetuximab-IRDye800. Neurooncol. 2018, 139, 135–143.
This manuscript is a resubmission of an earlier submission. The following is a list of the peer review reports and author responses from that submission.
Round 1
Reviewer 1 Report
Near-infrared molecular imaging of glioblastoma by Miltuximab-IRDye800CW as a potential tool for fluorescence-guided surgery.
The manuscript details the in vitro and in vivo application of an antibody (Miltuximab) targeting the proteoglycan Glypican-1. The antibody is attached to a NIR dye as a potential fluorescent tool for fluorescent guided surgery of glioblastoma.
The authors claim that the use of IRDye800 is superior to 5-ALA due to better tumour to background ratio, and that coupled with Miltuximab to target glypican-1, which is preferentially expressed on glioma cells, there is enhanced uptake by glioblastoma cells compared to control cells and control antibody-dye conjugate.
The manuscript is well written – a good standard of English throughout. The experiments are largely conducted well with appropriate controls, but there are some areas that have been overlooked or data lacking.
Main concern
Lack of brain in the experiments. The authors have tried to address this issue in the discussion, but the main concern here is that the brain has not been considered in either 1. Orthotopic model of brain tumour (which is superior to heterotopic) and/or 2. The distribution profile of the Miltuximab-IR800 conjugate (Figure 5). I find this unusual since the authors are focussing on brain tumours as their choice of tumour and have claimed that IR800 affords cm deep optical imaging penetration depth. If the authors have the images of the brain post injection and post mortem (Figure 5) then they must include this in the Figure, at least to show that their aspirations for moving into orthotopic animal models of glioblastoma has some strong foundation. The authors do not consider whether the conjugate has the appropriate properties to even gain access to the brain (e.g. size, charge, lipophilicity) nor cited any papers that might have alluded to this.
Another flaw in the experimental design is the lack of use of 5-ALA, since you refer to this molecule as being inferior to the conjugate. I understand this has been done in other publications, but it would have been more beneficial if a direct comparison had been made in your study, particularly of the TBR.
Minor concern
With regards to the characterisation of the Miltuximab-IR800, I’m not sure where the authors get 90% Multuximab-IR800 conjugate from. They cite that from size exclusion chromatography, followed by HPLC, 90% molecules were dye-labelled, but 90.2% monomeric for Miltuximab and 81.3% for Miltuximab-IR800. These percentages need some further explanation.
For the conjugation and characterization of Miltuximab-IR800, more details are required regarding the separation procedure via the spin column. Also, was a calibration curve used to determine the overall ratio? If so, these details need to be added.
In Figure 2- should the Y axis be absorbance rather than fluorescence?
On Page 4 line 131, should 8 days be changed to 9 days to match the Figure (4).
In the discussion (line 210-213), the authors state that 5-ALA is problematic in non-malignant tissues in cases of radiation necrosis or neurodegenerative diseases due to autofluorescence, but glypican-1 expression has also been shown to play a potential role in the pathogenesis of some of these diseases and Miltuximab specifically binds to glypican-1. Perhaps the authors could be a bit more balanced with their discussion regarding the benefits of Miltuximab-IR800 superiority over 5-ALA.
Typos
Line 39- remove extra space between ‘GBM’ and ‘has’
Line 99- this should be rephrased, as size exclusion chromatography is different to high performance liquid chromatography. Same with line 104. It also specifically states in figure 2 that SEC was the method used.
Line 300- I think should read 5 x 105.
Author Response
Dear Reviewer,
Thank you for reviewing our manuscript submitted for publication in Cancers. We appreciate your constructive suggestions. We have revised this manuscript and addressed your comments below. The comments are highlighted in italics, followed by our reply and amendments in regular font.
Comment:
Lack of brain in the experiments. The authors have tried to address this issue in the discussion, but the main concern here is that the brain has not been considered in either 1. Orthotopic model of brain tumour (which is superior to heterotopic) and/or 2. The distribution profile of the Miltuximab-IR800 conjugate (Figure 5). I find this unusual since the authors are focusing on brain tumours as their choice of tumour and have claimed that IR800 affords cm deep optical imaging penetration depth. If the authors have the images of the brain post injection and post mortem (Figure 5) then they must include this in the Figure, at least to show that their aspirations for moving into orthotopic animal models of glioblastoma has some strong foundation. The authors do not consider whether the conjugate has the appropriate properties to even gain access to the brain (e.g. size, charge, lipophilicity) nor cited any papers that might have alluded to this.
Reply:
Validation of Miltuximab-IR800 in an orthotopic glioma model is a necessary next step of our work. In this manuscript, we investigated the behavior of Miltuximab®- IR800 in vivo and its specific accumulation in GPC-1-expressing tumors following intravenous administration. As we aimed to investigate the accumulation and retention of the conjugate in the tumor over extended periods, we chose subcutaneous glioma model, which is well tolerated by the animals.
No fluorescence signal from the brain was detected in vivo. Accumulation of Miltuximab®-IR800 in healthy brain is unlikely due to the intact blood-brain barrier (BBB), and hence no images of the brain post mortem were collected from the animals. The lack of significant non-specific accumulation of Nimotuzumab-IR800 and control IgG-IR800 in the mouse brain has been reported [1], and no significant specific accumulation of Cetuximab-IR800 was found in the Cynomolgus Macaque brain [2]. The accumulation of the conjugate in the brain was significantly lower than that in the other critical organs.
On the other hand, Miltuximab®-IR800 is expected to accumulate in the brain tumours due to damaged blood-brain barrier and remain there due to the binding to Glypican-1. The conjugates of IR800 with integrin-targeting peptide or Cetuximab have been shown to gain access the orthotopic glioblastoma xenografts in mice [3] with no accumulation in the normal brain tissue. The accumulation of Cetuximab-IR800 in contrast-enhancing glioma was later demonstrated in patients [4]. Similarly to Cetuximab, Miltuximab® is a chimeric mouse/human IgG1 antibody. Miltuximab®-IR800 is therefore expected to behave similarly in vivo with accumulation in contrast-enhancing glioma, but not in normal brain tissue.
Amendment:
The following sentences are added to the Introduction section of the manuscript:
“The conjugate of IR800 with anti-EGFR antibody Cetuximab has been shown to access contrast-enhancing brain tumors and bind to glioma cells expressing target antigen without accumulation in the normal brain tissue in mice [3] and patients [4]. The lack of significant accumulation of such IR800-antibody conjugates in the brain has been also shown in mice [1], and Cynomolgus Macaques [2].”
“Similarly to Cetuximab-IR800, Miltuximab®-IR800 is expected to accumulate in contrast-enhancing brain tumors with damaged blood brain barrier without accumulation in healthy brain tissue.”
Comment:
Another flaw in the experimental design is the lack of use of 5-ALA, since you refer to this molecule as being inferior to the conjugate. I understand this has been done in other publications, but it would have been more beneficial if a direct comparison had been made in your study, particularly of the TBR.
Reply:
The authors agree that the superiority of Miltuximab®-IR800 over 5-ALA cannot be established without further testing and direct comparison. We therefore made the following changes in the discussion section.
Amendment:
The original paragraph
“Near-infrared molecular imaging has the potential to overcome both limitations of 5-ALA. The agents designed for molecular imaging usually consist of a targeting component (cancer-specific antibody or antibody fragment) and an imaging component (fluorophore). The presence of a targeting molecule has the potential to provide better specificity than passive accumulation or 5-ALA. The NIR fluorophore IR800 affords centimeter-deep optical imaging penetration depth on the background of diminished autofluorescence, and as such is much more advantageous in comparison with ultraviolet-visible fluorophores such as 5-ALA or fluorescein [41].”
is rewritten to read:
“Near-infrared molecular imaging has the potential to overcome both limitations of 5-ALA. The agents designed for molecular imaging usually consist of a targeting component (cancer-specific antibody or antibody fragment) and an imaging component (fluorophore). The NIR fluorophore IR800 affords centimeter-deep optical imaging penetration depth on the background of diminished autofluorescence, and as such is more advantageous in comparison with ultraviolet-visible fluorophores such as 5-ALA or fluorescein [43]. The presence of a targeting molecule has the potential to provide better specificity than passive accumulation or 5-ALA. Indeed, monoclonal antibodies are exquisitely specific for target antigen, and, given the established expression of GPC-1 in malignancy, including GBM, and the lack of GPC-1 expression seen in normal tissue, molecular targeting using Miltuximab® may afford a specificity advantage over the use of 5-ALA. Of course, access of systemically delivered drugs to the brain remains a challenge for drug delivery in GBM – 5-ALA as a small molecule achieves this, and, while antibodies have been known to cross the blood-brain barrier in GBM patients, access may not be equivalent to 5-ALA – a question that must be addressed to establish the utility of Miltuximab®-IR800 for imaging of GBM.”
Comment:
With regards to the characterisation of the Miltuximab-IR800, I’m not sure where the authors get 90% Multuximab-IR800 conjugate from. They cite that from size exclusion chromatography, followed by HPLC, 90% molecules were dye-labelled, but 90.2% monomeric for Miltuximab and 81.3% for Miltuximab-IR800. These percentages need some further explanation.
Reply:
The authors agree with the reviewer’s critique regarding this ambiguous interpretation.
Amendment:
In the section 2.2. Characterization of Miltuximab®-IR800, the text
“The molecular purity was evaluated by size exclusion chromatography high-performance liquid chromatography (SEC-HPLC) and found to be 90.2% monomeric for Miltuximab® and 81.3% for Miltuximab®-IR800. This indicated that approximately 90% of Miltuximab® molecules were dye-labelled. Following the conjugation, the SEC-HPLC data also demonstrated a shift of the main antibody peak and lower retention time, consistent with an increase in the size of the resultant molecule due to the attachment of the dye (Figure 2)”
was replaced by
“The molecular purity was evaluated by size exclusion high-performance liquid chromatography (SEC-HPLC) and found to be 90.2% monomeric for Miltuximab® and 81.3% for Miltuximab®-IR800. Following the conjugation, the SEC-HPLC data also demonstrated a shift of the main antibody peak and lower retention time, consistent with an increase in the size of the resultant molecule due to the attachment of the dye (Figure 2). Together, these findings are consistent with approximately 90% of monomeric Miltuximab® molecules being dye-labelled (and increasing in size), and 10% of Miltuximab® molecules remaining unlabeled and detected as a lower-molecular-weight fraction of Miltuximab®-IR800 at figure 2.”
In section 4.3. Conjugation and characterization of Miltuximab®-IR800, the paragraph
“Analysis of the purity of the antibody and conjugates was performed by SEC-HPLC on an Agilent Bio SEC-3 column with UV detection at 280 nm, flow rate of 0.3 mL/min and column temperature of 25 ºC. A protein standard mixture (Bio-Rad, USA) was run as control prior to analysis of all other samples.”
was replaced by
The % purity of the antibody and conjugate samples were assessed by SEC using an Agilent Bio SEC-3 column on an Agilent 1200 HPLC system (Agilent Technologies, Inc., Santa Clara, USA). 5uL sample was injected via the autosampler of the HPLC system, chromatograph was recorded with UV detection at 280 nm, flow rate of 0.3 mL/min for 20 mins and column temperature of 25 ºC. A protein gel filtration standard mixture (Bio-Rad Cat#1511901, USA) was run as a system control prior to analysis of all other samples. Data processing was performed via Agilent ChemStation Software system (Agilent Technologies, Inc., Santa Clara, USA) which gave the % purity via integration of area-under-curve of individual peak.
Comment:
For the conjugation and characterization of Miltuximab-IR800, more details are required regarding the separation procedure via the spin column. Also, was a calibration curve used to determine the overall ratio? If so, these details need to be added.
Amendment:
More details regarding the separation and characterization of the conjugate. Section 4.3. Conjugation and characterization of Miltuximab®-IR800 now reads:
“The Miltuximab®-IR800 conjugate was subsequently separated from free dye by centrifugation at 1,000 × g for 2 minutes in a Zeba Spin Desalting Column with 7 kDa molecular weight cut off (Thermo Fisher Scientific, USA), supplied in the kit. To determine the final concentration of the conjugate and the dye/antibody molar ratio, absorbance spectra at 280 nm (A280) and 780 nm (A780) were determined by a microplate reader Pherastar (BMG Labtech, Germany). Following the protocol supplied with the kit, the dye/antibody molar ratio was calculated as [A780/εIR800]/[(A280 - (0.03 × A780))/εAntibody], where 0.03 is a correction factor for the absorbance of IR800 at 280 nm, the molar extinction coefficient of the dye (εIR800) is 270,000 M-1cm-1 and the molar coefficient of the protein (εAntibody) is 203,000 M-1cm-1. The final concentration of the conjugate in mg/ml was calculated as [(A280 - (0.03 × A780))/εAntibody] × MWAntibody × dilution factor, where MWAntibody is the antibody molecular weight. A human IgG Isotype Control (cat # 31154, Thermo Fisher Scientific, USA) was also conjugated with IR800 and the resulting control IgG-IR800 conjugate was characterized using the same method as for Miltuximab®-IR800.
Comment:
In Figure 2- should the Y axis be absorbance rather than fluorescence?
On Page 4 line 131, should 8 days be changed to 9 days to match the Figure (4).
Reply:
Thank you, the figure and text have been amended.
Comment:
In the discussion (line 210-213), the authors state that 5-ALA is problematic in non-malignant tissues in cases of radiation necrosis or neurodegenerative diseases due to autofluorescence, but glypican-1 expression has also been shown to play a potential role in the pathogenesis of some of these diseases and Miltuximab specifically binds to glypican-1. Perhaps the authors could be a bit more balanced with their discussion regarding the benefits of Miltuximab-IR800 superiority over 5-ALA.
Reply:
The authors agree that further testing of Miltuximab-IR800 and its comparison with 5-ALA are needed to substantiate the superiority of one agent over the other.
Amendment:
The original paragraph:
“Near-infrared molecular imaging has the potential to overcome both limitations of 5-ALA. The agents designed for molecular imaging usually consist of a targeting component (cancer-specific antibody or antibody fragment) and an imaging component (fluorophore). The presence of a targeting molecule has the potential to provide better specificity than passive accumulation or 5-ALA. The NIR fluorophore IR800 affords centimeter-deep optical imaging penetration depth on the background of diminished autofluorescence, and as such is much more advantageous in comparison with ultraviolet-visible fluorophores such as 5-ALA or fluorescein [41].”
is rewritten as
“Near-infrared molecular imaging has the potential to overcome both limitations of 5-ALA. The agents designed for molecular imaging usually consist of a targeting component (cancer-specific antibody or antibody fragment) and an imaging component (fluorophore). The NIR fluorophore IR800 affords centimeter-deep optical imaging penetration depth on the background of diminished autofluorescence, and as such is much more advantageous in comparison with ultraviolet-visible fluorophores such as 5-ALA or fluorescein [43].The presence of a targeting molecule has the potential to provide better specificity than passive accumulation or 5-ALA. Indeed, monoclonal antibodies are exquisitely specific for target antigen, and, given the established expression of GPC-1 in malignancy, including GBM, and the lack of GPC-1 expression seen in normal tissue, molecular targeting using Miltuximab® may afford a specificity advantage over the use of 5-ALA. Of course, access of systemically delivered drugs to the brain remains a challenge for drug delivery in GBM – 5-ALA as a small molecule achieves this, and, while antibodies have been known to cross the blood brain barrier in GBM patients, access may not be equivalent to 5-ALA – a question that must be addressed to establish the utility of Miltuximab®-IR800 for imaging of GBM.”
Comment:
Typos
Line 39- remove extra space between ‘GBM’ and ‘has’
Line 99- this should be rephrased, as size exclusion chromatography is different to high performance liquid chromatography. Same with line 104. It also specifically states in figure 2 that SEC was the method used.
Line 300- I think should read 5 x 105.
Reply:
The figure and text are amended.
References:
Bernhard, W.; El-Sayed, A.; Barreto, K.; Gonzalez, C.; Hill, W.; Parada, A.C.; Fonge, H.; Geyer, C.R. Near infrared fluorescence imaging of EGFR expression in vivo using IRDye800CW-nimotuzumab. Oncotarget 2018. Zinn, K.R.; Korb, M.; Samuel, S.; Warram, J.M.; Dion, D.; Killingsworth, C.; Fan, J.; Schoeb, T.; Strong, T. V.; Rosenthal, E.L. IND-Directed Safety and Biodistribution Study of Intravenously Injected Cetuximab-IRDye800 in Cynomolgus Macaques. Mol. Imaging Biol. 2014. Warram, J.M.; De Boer, E.; Korb, M.; Hartman, Y.; Kovar, J.; Markert, J.M.; Gillespie, G.Y.; Rosenthal, E.L. Fluorescence-guided resection of experimental malignant glioma using cetuximab-IRDye 800CW. Br. J. Neurosurg. 2015. Miller, S.E.; Tummers, W.S.; Teraphongphom, N.; van den Berg, N.S.; Hasan, A.; Ertsey, R.D.; Nagpal, S.; Recht, L.D.; Plowey, E.D.; Vogel, H.; et al. First-in-human intraoperative near-infrared fluorescence imaging of glioblastoma using cetuximab-IRDye800. J. Neurooncol. 2018, 139, 135–143.
Reviewer 2 Report
This manuscript entitled “Near-infrared molecular imaging of glioblastoma by Miltuximab-IRDye800CW as a potential tool for fluorescence-guided surgery” shows that anti-Glypican-1 antibody, Miltuximab conjugated with IRDye800CW accumulates specifically in glioblastoma cells in the subcutaneous implantation model. This article is considered of value in that a novel molecular imaging agent which may have some advantages compared to 5-ALA is presented. However, this reviewer has two questions before recommendation for publication;
1) Figure 5
No data of brain is indicated. Extremely, not subcutaneous but intracerebral implantation model is more preferable because this study aims intraoperative imaging in neurosurgery. At least, ex vivo fluorescence of brain should be checked.
2) Figures 4 and 6
Only data using U87 are indicated. This article is single cell study in these key experiments. Data of U251 (or more cell lines) are desired.
Author Response
Dear Reviewer,
Thank you for reviewing our manuscript submitted for publication in Cancers. We appreciate your constructive suggestions. We have revised this manuscript and addressed your comments below. The comments are highlighted in italics, followed by our reply and amendments in regular font.
Comment:
Figure 5
No data of brain is indicated. Extremely, not subcutaneous but intracerebral implantation model is more preferable because this study aims intraoperative imaging in neurosurgery. At least, ex vivo fluorescence of brain should be checked.
Reply:
Validation of Miltuximab-IR800 in an orthotopic glioma model is a necessary next step of our work. In this manuscript, we investigated the behaviour of Miltuximab®- IR800 in vivo and its specific accumulation in GPC-1-expressing tumors following intravenous administration. As we aimed to investigate the accumulation and retention of the conjugate in the tumour over extended periods, we chose subcutaneous glioma model as it is well tolerated by the animals.
No fluorescence signal from the brain was detected in vivo. Accumulation of Miltuximab®-IR800 in healthy brain is unlikely due to the intact blood-brain barrier (BBB), and hence no images of the brain post mortem were collected from the animals. The lack of significant non-specific accumulation of Nimotuzumab-IR800 and control IgG-IR800 in the mouse brain has been reported [1], and no significant specific accumulation of Cetuximab-IR800 was found in the Cynomolgus Macaque brain [2]. The accumulation of the conjugate in the brain was significantly lower than that in the other critical organs.
On the other hand, Miltuximab®-IR800 is expected to accumulate in brain tumours due to damaged blood-brain barrier and remain there due to the binding to Glypican-1. The conjugates of IRDye800CW with integrin-targeting peptide or cetuximab have been shown to access orthotopic glioblastoma xenografts in mice [3] without accumulating in normal brain tissue. The accumulation of Cetuximab-IR800 in contrast-enhancing glioma was later demonstrated in patients [4]. Similarly to Cetuximab, Miltuximab® is a chimeric mouse/human IgG1 antibody. Miltuximab®-IR800 is therefore expected to behave similarly in vivo with accumulation in contrast-enhancing glioma, but not normal brain tissue.
Amendment:
The following sentences have been added to the Introduction section of the manuscript:
“The conjugate of IR800 with anti-EGFR antibody Cetuximab has been shown to gain access contrast-enhancing brain tumors and bind to glioma cells expressing target antigen without accumulation in the normal brain tissue in mice [3] and patients [4]. The lack of significant accumulation of such IR800-antibody conjugates in the brain was also shown in mice [1], and Cynomolgus Macaques [2].”
“Similarly to Cetuximab-IR800, Miltuximab®-IR800 is expected to accumulate in contrast-enhancing brain tumors with damaged blood brain barrier without accumulation in healthy brain tissue.”
Comment:
Figures 4 and 6
Only data using U87 are indicated. This article is single cell study in these key experiments. Data of U251 (or more cell lines) are desired.
Reply:
We demonstrated Glypican-1 expression in both U-87 and U-251 cell lines. Both cell lines have been previously shown to form xenografts accessible to IR800-antibody conjugate. In order to reduce the number of used animals, we selected lower-expressing U-87 cells for the animal model. U-251 cells have significantly higher expression of Glypican-1 and therefore are unlikely to have less accumulation of Miltuximab®-IR800.
References:
Bernhard, W.; El-Sayed, A.; Barreto, K.; Gonzalez, C.; Hill, W.; Parada, A.C.; Fonge, H.; Geyer, C.R. Near infrared fluorescence imaging of EGFR expression in vivo using IRDye800CW-nimotuzumab. Oncotarget 2018. Zinn, K.R.; Korb, M.; Samuel, S.; Warram, J.M.; Dion, D.; Killingsworth, C.; Fan, J.; Schoeb, T.; Strong, T. V.; Rosenthal, E.L. IND-Directed Safety and Biodistribution Study of Intravenously Injected Cetuximab-IRDye800 in Cynomolgus Macaques. Mol. Imaging Biol. 2014. Warram, J.M.; De Boer, E.; Korb, M.; Hartman, Y.; Kovar, J.; Markert, J.M.; Gillespie, G.Y.; Rosenthal, E.L. Fluorescence-guided resection of experimental malignant glioma using cetuximab-IRDye 800CW. Br. J. Neurosurg. 2015. Miller, S.E.; Tummers, W.S.; Teraphongphom, N.; van den Berg, N.S.; Hasan, A.; Ertsey, R.D.; Nagpal, S.; Recht, L.D.; Plowey, E.D.; Vogel, H.; et al. First-in-human intraoperative near-infrared fluorescence imaging of glioblastoma using cetuximab-IRDye800. J. Neurooncol. 2018, 139, 135–143.
Reviewer 3 Report
In this manuscript Polikarpov et al. suggest an interesting method to highlight the region affected by the tumor through a fluorescence-guided surgery.
The identification of the target and method described are fine and this tool can really improve GBM surgery rendering it more precise than the actual use of 5-ALA.
Author Response
Dear Reviewer,
Thank you for reviewing our manuscript for publication in Cancers. We appreciate your feedback.
Round 2
Reviewer 2 Report
Figure 5
This reviewer could accept the authors’ mention.
Figures 4 and 6
Only expression of glypican-1 (detected by MIL-38) was checked about U251 in Figure 1. As this study aims to suggest clinical usability of Miltuximab®-IR800 in surgery, it is important to show in vivo detection of tumor by Miltuximab®-IR800. This reviewer thinks that at least two or three cell lines should be checked in the “key” experiments.
However, this reviewer would entrust the editor to decide whether the revised manuscript is accepted in present form or not.